# A Protein Asteroid with PIN Domain in Silkworm *Bombyx mori* Is Involved in Anti-BmNPV Infection

**DOI:** 10.3390/insects14060550

**Published:** 2023-06-13

**Authors:** Yuchen Xia, Mouzhen Jiang, Xiaoxuan Hu, Qing Wang, Cen Qian, Baojian Zhu, Guoqing Wei, Lei Wang

**Affiliations:** School of Life Science, Anhui Agricultural University, Hefei 230036, China; rose_judas@foxmail.com (Y.X.); jiangmouzhen@stu.ahau.edu.cn (M.J.); qiancenqiancen@163.com (C.Q.); zhubaojian@ahau.edu.cn (B.Z.);

**Keywords:** *Bombyx mori*, protein asteroid, nuclease, PIN domain, NPV

## Abstract

**Simple Summary:**

Nucleases are highly diverse enzymes that cleave phosphodiester bonds of nucleic acids with highly diverse enzymes. A widespread superfamily of nucleases contains PIN-like domains, which are identified in eukaryotes and prokaryotes. The domesticated silkworm *Bombyx mori*, as a lepidoptera model insect, has numerous advantages in life science and can be used as an important model in various research areas. However, few nucleases on RNA insensitivity or virus defense have been studied in *B. mori*. In this study, we identified a protein asteroid (BmAst) in silkworm *B. mori*, containing the PIN domain and XPG domain. *BmAst* gene was highest expressed in hemocytes and fat body of the 5th larvae. The *BmAst* gene expression was significantly induced by *Bombyx mori* nucleopolyhedrovirus (BmNPV) or dsRNA. By interference with *BmAst* gene expression, the proliferation of BmNPV in *B. mori* was significantly increased, whereas the survival rate of silkworm larvae was significantly lower when compared with the control. Our results indicate that BmAst is involved in silkworm resistance to BmNPV infection.

**Abstract:**

Nuclease is a type of protein that degrades nucleic acids, which plays an important role in biological processes, including RNA interference efficiency and antiviral immunity. However, no evidence of a link between nuclease and *Bombyx mori* nucleopolyhedrovirus (BmNPV) infection in silkworm *B. mori* has been found. In this study, a protein asteroid (BmAst) containing the PIN domain and XPG domain was identified in silkworm *B. mori*. *BmAst* gene was highest expressed in hemocytes and fat body of the 5th instar larvae, and high expression in the pupa stage. The transcriptional levels of the *BmAst* gene in 5th instar larvae were significantly induced by BmNPV or dsRNA. After knocking down *BmAst* gene expression by specific dsRNA, the proliferation of BmNPV in *B. mori* was increased significantly, whereas the survival rate of larvae was significantly lower when compared with the control. Our findings indicate that BmAst is involved in silkworm resistance to BmNPV infection.

## 1. Introduction

The metabolism of nucleic acids plays an important role in biological processes in all kingdoms of life. Nucleases are highly diverse enzymes that cleave phosphodiester bonds of nucleic acids with highly diverse enzymes [1]. DNA/RNA nonspecific nucleases have a broad range of substrate specificity. Within this nuclease family, dsRNA, ssRNA, ssDNA, dsDNA, and DNA/RNA mixtures can all be degraded, but nuclease activity is usually highest when dsRNA is the substrate [2]. Some dsRNases belonging to the DNA/RNA nonspecific nuclease group have been found in insects. Studies have found Bm-dsRNase in the gut of silkworms, Sg-dsRNase in desert locusts, and Ld-dsRNase in the Colorado potato beetle, to name a few [3,4,5]. dsRNases were reported in limited RNAi efficiency in many insects since they destroy dsRNA before dsRNA from being internalized by cells where the RNAi mechanism takes place [6,7]. Nuclease activity decreasing the RNAi responses have been reported in some insects, such as the sweet potato weevil *Cylas puncticollis* [8], Colorado potato beetle *Leptinotarsa decemlineata* [5], tobacco cutworm, and *Spodoptera litura* [9].

PIN-like domains, annotated type IV pili twitching motility protein (PilT N-terminal domain, PIN), constitute a widespread superfamily of nucleases [1]. PIN domains were first identified in bacteria and found to have ribonuclear activity in antitoxin systems [10]. Then some proteins with PIN domains were identified in additional eukaryotes and prokaryotes. The crystal structures of archaeal PIN domains show homology with several exonucleases, including T4 phage RNase H and flap endonuclease (FEN1), suggesting Mg^2+^-dependent exonucleases [11]. The PIN domains in eukaryotes are associated with the nonsense-mediated decay of RNA [12]. A lepidopteran-specific nuclease gene, REase in Asian corn borer *Ostrinia furnacalis*, contains a PIN domain at the 5′-end and encodes a nuclease that contributes to RNAi insensitivity in lepidoptera insect order [13].

The domesticated silkworm *Bombyx mori*, as a lepidoptera model insect, has numerous advantages in life science and can be used as an important model in various research areas [14,15]. Genetic manipulation technologies have been extensively applied in *B. mori* gene functional analysis and the development of silkworms as bioreactors [16]. However, *Bombyx mori* nucleopolyhedrovirus (BmNPV) is a major pathogen in silkworm breeding that causes serious economic loss to the sericulture industry worldwide [17]. BmNPV is a typical baculovirus belonging to Group I, with glycoprotein 64 as the main envelope protein [18]. Improving the resistance of silkworms to BmNPV has important practical significance for sericulture. Some antiviral genes in *B. mori* have been reported against BmNPV infection by overexpression in transgenic silkworms, such as *PEPCK*, *hycu-ep32*, *Bmlipase-1*, and *BmHsp19.9* [19,20,21,22]. Additionally, knocking down or out of some viral genes has been studied to enhance silkworm resistance to BmNPV, including *IE-0* and *IE-2*, *lef8* and *lef9*, *BmFerHCH*, and *BmATG13* [23,24,25,26]. The regulations of the silkworm host immune pathway also are related to the silkworm against BmNPV. Some host signaling pathways, such as MAPK signal pathways, are important in NPV infection [27]. Spry is a negative regulator upstream of ERK signaling, the viral content and mortality in Spry-I (transgenic RNAi line of BmSpry) were significantly higher than those in control after infection with the BmNPV [28]. However, the molecular basis of silkworm resistance to BmNPV has not yet been fully understood.

Several techniques have been developed as a result of research on the interaction between viruses and the siRNA pathway, including methods for detecting hidden viruses in insect cell lines, improving baculovirus expression systems, and the potential application of virus-induced gene silencing (VIGS) for the control of pests and diseases [29]. RNAi pathway as innate antiviral immunity has been reported, and nuclease and RNAi efficiency are closely related [30]. In this study, a protein asteroid in *B. mori* (BmAst) was identified as a nuclease belonging to the PIN superfamily. The gene expressions of *BmAst* were analyzed in the *B. mori* larvae treated by dsRNA or BmNPV. The relationship between *BmAst* expression and proliferation of BmNPV in *B. mori* was measured by RNA interference. Our results showed that *BmAst* gene expressions were significantly induced by BmNPV or dsRNA. After knocking down the *BmAst* gene, the proliferation of BmNPV in *B. mori* was significantly increased, whereas the survival rate of silkworm larvae was significantly lower compared with the control. Our results suggest that BmAst has a role in silkworm resistance to BmNPV infection.

## 2. Materials and Methods

### 2.1. Insect Rearing and Microorganism

The experimental *Bombyx mori* (Nistari strain) was maintained in our laboratory. The *B. mori* larvae were reared with fresh mulberry leaves at 25 ± 1 °C under a 12L:12D photoperiod and 60 ± 10% relative humidity. BmNPV T3 strain, bacterial (*Escherichia coli*, *Bacillus thuringiensis*), or fungus (*Beauveria bassiana*) were stored at −20 °C in our laboratory and purified according to the protocol reported [31].

### 2.2. Sequence Analysis of BmAst Gene

Total RNA was isolated from *B. mori* tissues using a TRIzol reagent (Transgen, Beijing, China) and reverse-transcribed into complementary DNA (cDNA) using a TransScript^®^ First-Strand cDNA Synthesis SuperMix (Transgen) according to the manufacturer’s instructions. Based on the sequence at GenBank (GenBank No. XM_021350017.2), a pair of primers (BmAst-F and BmAst-R) (Table 1) was designed to amplify the full-length ORF of the *BmAst* gene. The aimed PCR band was purified, cloned using a pEASY^®^-T1 Cloning Kit (Transgen), and then sequenced by Sangon Biotech. The conserved domains of predicted BmAst protein were searched at the NCBI (http://blast.ncbi.nlm.nih.gov/Blast.cgi (accessed on 11 June 2023)) and Pfam (http://pfam.xfam.org/ (accessed on 11 June 2023)). Homologous sequences of BmAst protein in insects were obtained from the NCBI. The phylogenetic tree was constructed using the neighbor-joining method by MEGA 7 and was tested for reliability using 1000 bootstrap repetitions [32].

### 2.3. Quantitative Real-Time Polymerase Chain Reaction

Total RNA was extracted from different tissues of *B. mori* using a TRIzol reagent and reverse-transcribed into cDNA using ReverTra Ace^®^ qPCR RT Master Mix with gDNA Remover (TOYOBO, Shanghai, China). For quantitative real-time polymerase chain reaction (qRT-PCR) analysis, the diluted cDNAs (10×) were used as the template, and the *BmActin A3* gene (GenBank No. U49854) was used as the internal control gene [31]. qRT-PCR was performed by a CFX96TM real-time detection system (Bio-Rad, Hercules, CA, USA). The program was set as predenaturation at 94 °C for 30 s, followed by 40 cycles at 95 °C for 5 s, 57 °C for 15 s, and 72 °C for 10 s. The primers of test genes (Table 1) were designed by Primer 3 (https://primer3.ut.ee/ (accessed on 11 June 2023)) based on their known sequences. A melting curve (65–95 °C) was analyzed to test the unique and specific PCR product for each reaction. The result data were analyzed using the 2^−ΔΔCt^ method [33]. All the qRT-PCR experiments were repeated three times.

### 2.4. Tissue Distribution and Developmental Profiles of BmAst Gene

The day 3, fifth-instar larvae were dissected on ice to analyze the tissue expression of the *BmAst* gene in *B. mori* larvae. The head and six tissues (silk gland, hemocytes, fat body, midgut, epidermis, and Malpighian tubule) were collected. For the developmental stages expression analysis of *BmAst*, egg stages, the whole body of *B. mori* in five larval stages (first, second, third, fourth, and fifth instars), pupa, and moth were collected. The expressions of *BmAst* mRNA were measured by qRT-PCR.

### 2.5. Different Microbial Infections

*B. mori* larvae (fifth instar, first day) were injected with the heated-inactivated Gram-negative bacterium (*Escherichia coli*, 5 μL, 10^9^ cfu/mL), a Gram-positive bacterium (*Bacillus thuringiensis*, 5 μL, 10^9^ cfu/mL), fungus (*Beauveria bassiana*, 5 μL, 10^9^ cfu/mL), virus (BmNPV, 5 μL, 10^9^ occlusion bodies (OBs)/mL) or phosphate-buffered saline buffer (PBS, 5 μL, as negative control), respectively. The fat body of *B. mori* was collected at 1.5, 3, 6, 12, 24, and 48 h post-infection by dissecting the larvae. All the experiments were carried out in three biological replicates. The expressions of *BmAst* mRNA were measured by qRT-PCR. Because of the low expression of *BmActin A3* after microbial infection, the *B. mori glyceraldehyde-3-phosphate dehydrogenase* gene (*GAPDH*, GenBank No. ABA43638) was used as an internal control for microbial infection [31,34,35].

### 2.6. dsRNA Synthesis

dsRNAs of *BmAst* or EGFP were synthesized and purified by the T7 RiboMAX^TM^ Express RNAi System (Promega, Madison, WI, USA) according to the manufacturer’s instructions. The target gene-containing plasmids were used in the PCR procedure. The specific primers for *BmAst* or *EGFP* containing a T7 promoter sequence (Table 1) at the 5′-end were designed for in vitro dsRNA synthesis. The synthesis was analyzed with 1% agarose gel, and dsRNA concentrations were determined by a Nanodrop instrument (Thermo, Wilmington, NC, USA).

### 2.7. Expressions of BmAst Gene after dsEGFP Injection

*B. mori* larvae (fifth instar, third day) were injected with dsEGFP (4 μg, 5 μL) to test the dsRNA effect on *BmAst* gene expression. The PBS was injected as a control. After 4 h injection, the fat body was collected for analyzing *BmAst* expression by qRT-PCR. The RNAi core gene, *Ago-2* (GenBank No. NM_001043530) and *Dicer-2* (GenBank No. NM_001193614.1) in *B. mori*, which can be induced by dsEGFP, were analyzed.

### 2.8. RNAi Assays

*B. mori* larvae (first day of 5th instar) were injected with dsBmAst (5 μL, 20 μg), or dsEGFP (5 μL, 20 μg) as a control. After 24 h or 48 h of dsBmAst injections, the mRNA expression of *BmAst* in *B. mori* fat body was analyzed by qRT-PCR. The larvae were fed with BmNPV (5 μL, 10^9^ OBs/mL) on mulberry leaves after 48 h *dsBmAst* injection. *B. mori* larvae continued feeding with fresh mulberry leaves. The larvae challenged with BmNPV only served as controls. The mortality of silkworms was calculated after being infected with BmNPV. Each biological sample (30 larvae) was carried out in three replicates.

DNA from the *B. mori* fat body after 48 h BmNPV feeding was extracted using Ezup Column DNA Extraction Kit (Sangon Biotech, Shanghai, China) for analyzing BmNPV proliferation. DNA concentrations were tested by a NanoDrop instrument. The primers for the *VP39* gene (GenBank No. NC_001962.1) were used to determine BmNPV proliferation (estimating viral DNA) in *B. mori* by qPCR. Primer information used for qPCR is shown in Table 1. The amplification program of qPCR was set as follows: 94 °C for 30 s, 30 cycles of 94 °C for 5 s and 60 °C for 30 s, and 72 °C for 20 s. Relative BmNPV genome copy numbers were evaluated using a standard curve with VP39 copies in BmNPV. Each qPCR was three replicates, and each biological sample was three replicates.

### 2.9. Statistic Analysis

The data in this study were presented as the mean ± standard error. Data were analyzed using a one-way analysis of variance analysis (ANOVA) (Tukey’s test) by DPS software [36]. *p* < 0.05 was considered a significant difference (* *p* < 0.05 and ** *p* < 0.05).

## 3. Results

### 3.1. Sequence Analysis of BmAst Gene and Its Predicted Protein

After PCR amplification and sequencing verification, the protein asteroid gene sequence of *B. mori* (BmAst) contains an open reading frame of 2241 base pairs encoding a 746-residue polypeptide (GenBank No. XM_021350017.2). SignalP5.0 analysis shows there is no signal peptide in the BmAst protein, indicating that BmAst protein may not be secreted outside the cell and belongs to intracellular protein. The predicted molecular weight of the protein is 87.72 kDa, and the theoretical isoelectric point is 6.77. The conserved domain analysis shows that the protein contains a PIN (PilT N terminus) domain and an XPG (Xeroderma pigmentosum complementation group G) domain (Figure 1A). The phylogenetic tree reveals that the BmAst protein clustered with the protein asteroid of *Bombyx mandarina*, and lepidoptera insects branch separately from other species (Figure 1B).

### 3.2. Tissues Distribution and Developmental Expression of BmAst

Quantitative real-time PCR results show that the *BmAst* gene is widely expressed in all tested tissues of 3-day-old fifth-instar larvae, with the highest expression level in hemocytes and fat body and the lowest expression in the epidermis (Figure 2A). At various developmental stages, results show that the *BmAst* gene is expressed in all growth stages of silkworms, and the expression level is the highest in the pupa stage (Figure 2B). These results indicate that the *BmAst* gene could have one essential function that is necessary for all tissues and developmental stages of silkworms.

### 3.3. BmAst Genes Respond to Different Pathogenic Microorganisms

To study the expression level of the *BmAst* gene in the fat body in response to four different pathogenic microorganisms, fat body from silkworms is extracted at 1.5, 3, 6, 12, 24, and 48 h after infection, and the expression levels of *BmAst* are detected by qRT-PCR. The results show that the *BmAst* gene does not significantly respond to bacterial infection by *E. coli* (Gram-negative bacterium), *B. thuringiensis* (Gram-positive bacterium), and *B. bassiana* (Fungus) compared with the control group injected with PBS, and their expression level remain almost unchanged (Figure 3). When the silkworm is injected with BmNPV (virus), the transcriptional level of the *BmAst* gene begins to significantly increase from 24 to 48 h after infection (Figure 3). The *BmAst* gene is induced by BmNPV, which suggests that the *BmAst* gene may be involved in the immune response to BmNPV infection in silkworms.

### 3.4. The Effect of dsRNA on BmAst Expression

To examine the influence of dsRNA on *BmAst* expression in fat body, exogenous dsEGFP was injected into *B. mori* larvae. *Dicer-2* and *Ago-2* genes in the small interfering RNA (siRNA) pathway are also detected by qRT-PCR. The results show that exogenous dsRNA can significantly up-regulate the expression of *BmAst* compared with the control (Figure 4A). *Dicer-2* and *Ago-2* genes in *B. mori* are also significantly induced by dsEGFP (Figure 4B,C). *BmAst* gene has similar expression profiles with *Dicer-2* and *Ago-2* after dsRNA treatment.

### 3.5. Knocking down of BmAst Gene Enhances BmNPV Replication in Silkworm

To understand the effect of the *BmAst* gene on the BmNPV proliferation in silkworms, dsBmAst is injected into the day 1, fifth-instar larva. qRT-PCR results show that the expression level of the *BmAst* gene is significantly down-regulated after 48 h dsBmAst injection (Figure 5A). Next, BmNPV was fed to silkworm larvae. Then, the fat body was collected after 48 h BmNPV-feeding. DNA in the fat body was extracted for qPCR. Results show that the proliferation of BmNPV in the *BmAst* gene silencing group is significantly increased compared with the other two groups (Figure 5B). In addition, the survival rate of larvae in the *BmAst* gene silencing group is significantly lower than the other two groups from 5 to 7 days after BmNPV-feeding (Figure 5C).

## 4. Discussion

In this study, we reported the characteristic of a protein asteroid in *B. mori* (BmAst). It belongs to the PIN domain superfamily, which is one of the largest and most diverse nuclease superfamilies [1]. The PIN domain is a small protein domain identified by three strictly conserved acidic residues and belongs to a large nuclease superfamily. XPG is a member of the structure-specific, 5′ nuclease family. The domain search of the BmAst protein predicted it to be a nuclease. The PIN domain-containing protein has the function of nuclease enzymes reported in some species, such as *O. furnaclis* and *C. elegans*. REase with a PIN domain at the 5′-end in *O. furnaclis*, could degrade various types of nucleic acids, such as dsRNA, dsDNA, ssRNA, ssDNA, and plasmid [13]. The SMG6’s PIN domain showed degradation activity for single-stranded RNA in vitro [10]. Smg-5 with PIN domain protein in *C. elegans* is involved in regulating the persistence of RNA interference [37]. Nuclease can reduce RNAi efficiency by degrading long dsRNA or processed siRNA fragments [38] and can also accelerate the degradation of foreign invading nucleic acids [39].

Our study showed that the *BmAst* gene was up-regulated by dsRNA and BmNPV and had no obvious response after larvae were infected with a Gram-positive bacterium, Gram-negative bacterium, or a fungus. This suggests that BmAst may be involved in RNAi efficiency and the defense against BmNPV infections. The dsRNA stability in insects is connected to RNAi efficiency, but the RNAi efficiency was variable in different species. Knockdown of nuclease activity in the insect gut enhanced RNAi efficiency is variable in the Colorado potato beetle, *Leptinotarsa decemlineata*, but not in the desert locust, *Schistocerca gregaria* [5]. Nuclease REase in Asian corn borer was also stimulated by dsRNA, which was closely related to RNAi efficiency by digesting dsRNA before processing by Dicer [13]. Recent work has implicated RNAi pathways in the establishment of persistent virus infections and in the control of DNA virus replication. Virus infection of RNAi-deficient *Drosophila* is often associated with higher mortality rates [40]. Flock house virus (FHV) was of low virulence in WT flies, with 50% of infected flies surviving 15 days postinoculation (dpi). In the *dcr-2* null mutation *Drosophila*, inoculation with the same dose of FHV resulted in 60% mortality by 6 dpi and 95% by 15 dpi [41]. RNA silencing is an adaptive antiviral immune response in animal cells and has a conserved pathway in the plant and animal kingdoms [41].

By knocking down the expression of the *BmAst* gene in silkworms, the proliferation of BmNPV was significantly increased, and the mortality rate after BmNPV infection was also increased. In conclusion, BmAst could indeed inhibit the proliferation of the BmNPV virus, which plays a crucial role in antiviral infections. The *Dicer2* gene in the RNAi pathway was involved in the anti-BmNPV virus in *B. mori* [31]. The *Dicer2* gene, as an RNAi component, has also been reported to affect the replication of other DNA viruses, such as Invertebrate iridescent virus 6 (IIV-6) [42], *Helicoverpa armigera* single nucleopolyhedrovirus (HaSNPV) [43] and Autographa californica multiple nucleopolyhedrovirus (AcMNPV) [44]. In *D. melanogaster*, the RNA-silencing endonuclease Argonaute 2 mediates specific antiviral immunity [45]. dsRNA is known to serve as a potent pathogen-associated molecular pattern (PAMP) in insects and induces the activation of an antiviral response against RNA and DNA viruses in diverse eukaryotes [29]. RNA polymerase II (RNAPII) in *Drosophila* bidirectionally transcribed IIV-6 DNA genome-specific AT-rich regions to generate dsRNA. Both replicative and naked IIV-6 genomes triggered the production of dsRNA in *Drosophila* cells. So, the RNAPII complex recognized invading viral DNA to synthesize virus-derived dsRNA, which activated the antiviral siRNA pathway in *Drosophila* [46]. BmAst may be a possible link between the RNAi pathway and antiviral responses. BmNPV is a circular double-stranded superhelical DNA (dsDNA) virus belonging to the baculovirus family. Baculoviruses can only be targeted by the siRNA pathway to inhibit their transcripts, and this indirect action is evidently unable to stop viral replication. Baculovirus genes that are affected by siRNAs tend to be late genes, which means that the infection is already well-established when the siRNA pathway comes in [43]. Therefore, although the siRNA pathway plays a specific part in baculovirus resistance, it does not seem to be a primary defense mechanism [47].

Altogether, our results suggest that BmAst protein containing a PIN family domain plays an important role in BmNPV infection in *B. mori*. Our results provide a target gene to improve virus resistance in *B. mori*. However, how BmAst affects BmNPV replication and the interaction between BmAst and RNAi pathway was unclear. The mechanism of BmAst involvement in antiviral function by the RNAi pathway will be investigated in the future.

## Figures and Tables

**Figure 1 insects-14-00550-f001:**
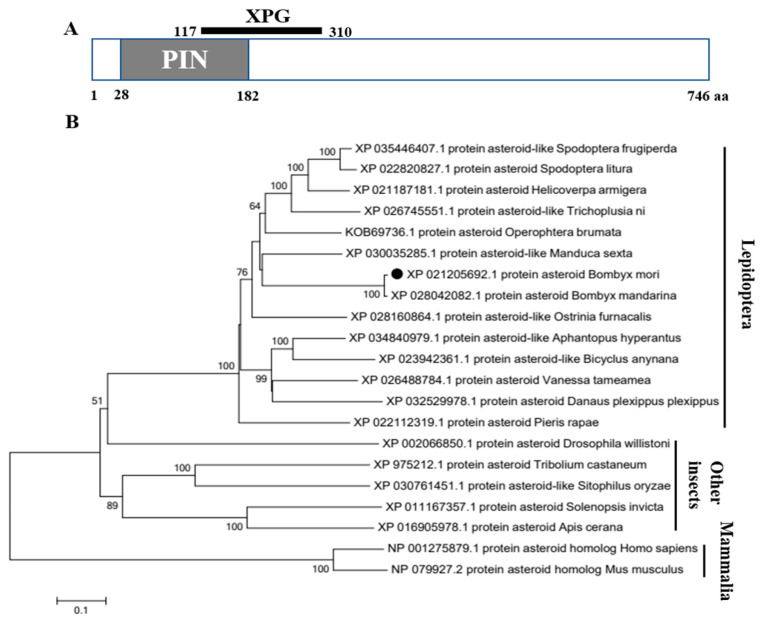
Bioinformatics analysis of *BmAst* gene and its predicted protein. (**A**) Pattern diagram of the conserved domain of BmAst amino acid sequence. PIN, PilT N terminus domain; XPG, Xeroderma pigmentosum complementation group G nuclease domain. (**B**) Phylogenetic analysis of BmAst from different species. The phylogenetic tree was constructed using the neighbor-joining algorithm by MEGA7, and the bootstrap (1000 repetitions) of the branches is indicated. The BmAst protein is marked with a black dot. The GenBank Accession No., gene name, and species name are indicated.

**Figure 2 insects-14-00550-f002:**
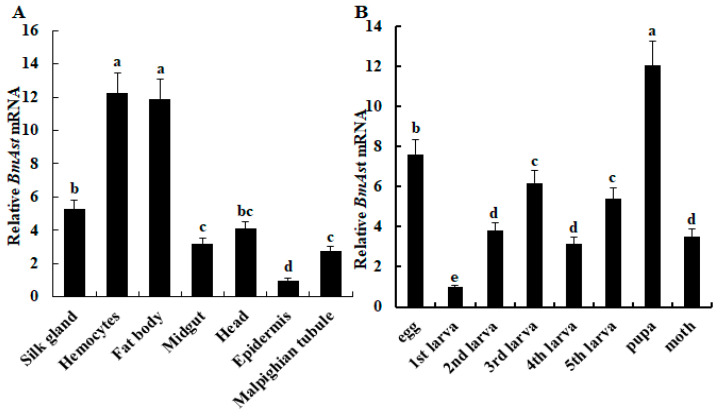
Expression patterns of the *BmAst* gene in larvae different tissues and developmental stages of *Bombyx mori.* (**A**) qRT-PCR analysis of *BmAst* mRNA expression in different tissues of 5th instar larvae. The *BmAst* mRNA level in the epidermis was used as the calibrator. (**B**) Expression profiles of *BmAst* mRNA during developmental stages. The *BmAst* mRNA level in the first-instar larvae was used as the calibrator. The bars show the mean ± SE (*n* = 3). Bars with different letter labels are significantly different (one-way ANOVA followed by Tukey’s test, *p* < 0.05).

**Figure 3 insects-14-00550-f003:**
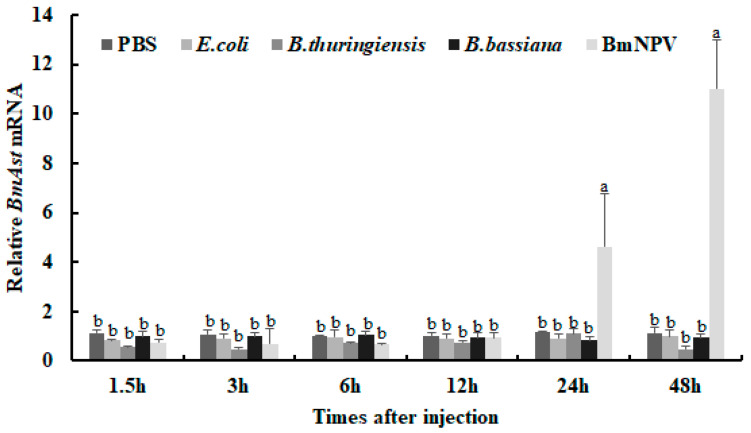
Expression patterns of the *BmAst* gene in the *B. mori* fat body of the larvae after microorganism injection. The 5th instar larvae of *B. mori* were injected with *E. coli*, *B. thuringiensis*, *B. bassiana*, or BmNPV. PBS was injected as a control. Fat body tissue of *B. mori* was collected at 1.5, 3, 6, 12, 24, and 48 h after injection. The *BmAst* mRNA after PBS-injected 6 h was designated as the calibrator. Bars represent mean ± SE (*n* = 3). Bars labeled with different letters are significantly different (one-way ANOVA followed by Tukey’s test, *p* < 0.05).

**Figure 4 insects-14-00550-f004:**
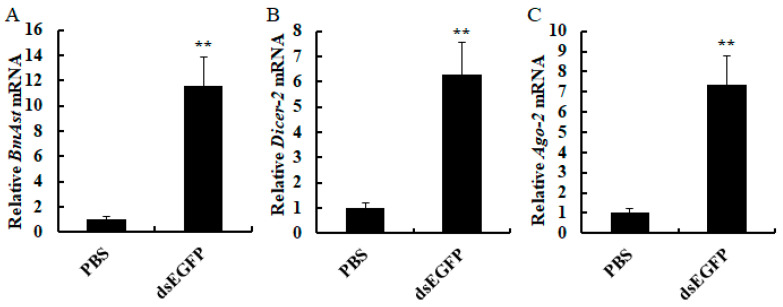
The expression levels of *BmAst* induced by dsEGFP. (**A**) The expression levels of *BmAst* after dsEGFP injection. (**B**) The expression levels of *BmDicer-2* after dsEGFP injection. (**C**) The expression levels of *BmAgo-2* after dsEGFP injection. Bars represent mean ± SE (*n* = 3). Bars labeled with double asterisks are extremely significantly different (Student’s *t*-test, ** *p* < 0.01).

**Figure 5 insects-14-00550-f005:**
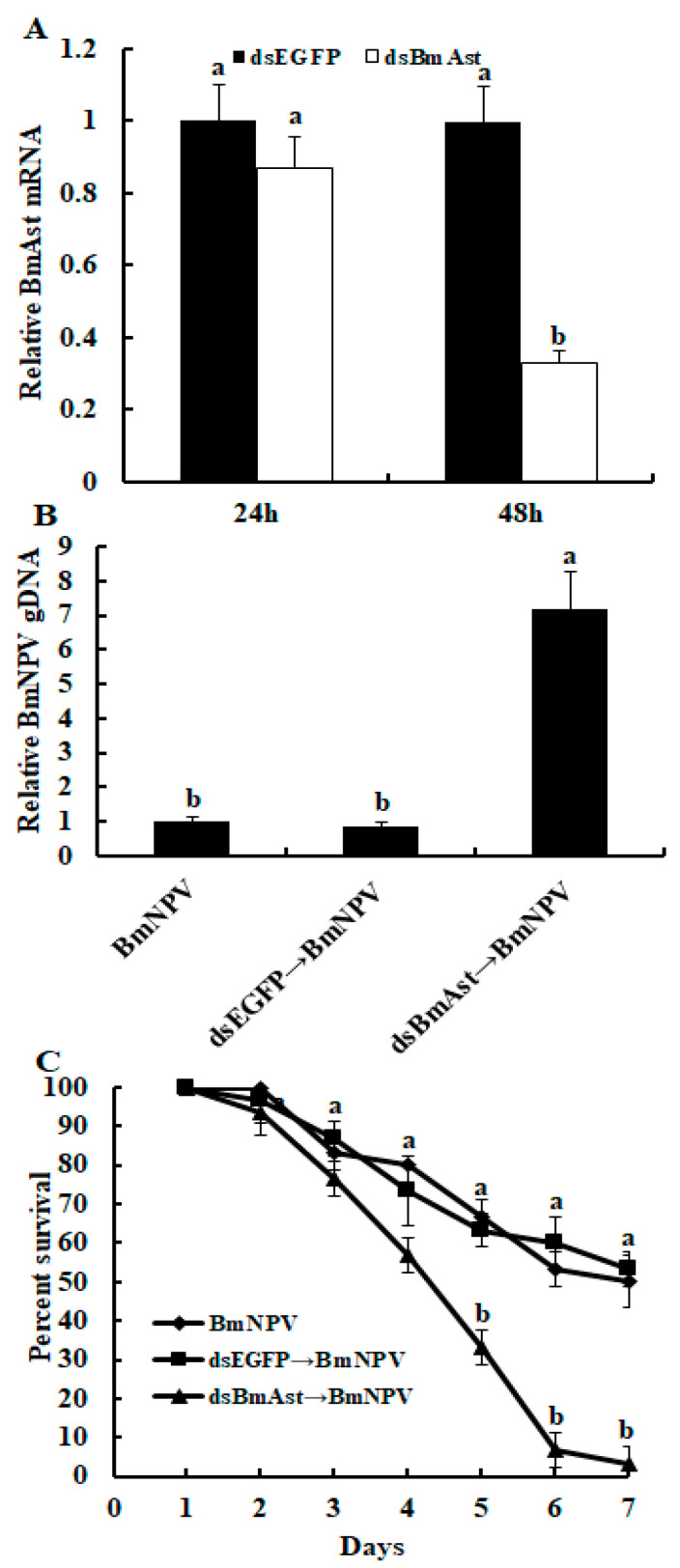
Effect of *BmAst* gene silencing on BmNPV proliferation rate and mortality in silkworm. (**A**) qRT-PCR analysis of the *BmAst* gene after dsBmAst injection. (**B**) BmNPV genomic DNA (gDNA) replication increased after *BmAst* gene silencing. DNA sample was extracted from the fat body at 48 h after BmNPV infection, and gDNA of BmNPV was analyzed by qPCR. (**C**) The survival rate of silkworms infected by BmNPV significantly decreased after the *BmAst* gene silencing. Bars represent mean ± SE (*n* = 3). Bars labeled with different letters are significantly different (one-way ANOVA followed by Tukey’s test, *p* < 0.05).

**Table 1 insects-14-00550-t001:** Sequences of primers used in this study.

Primers	Sequence (5′–3′)	Purpose
BmAst-F	CGGGATCCATGGGAGTACGAGGACTAACTACTT	Protein expression
BmAst-R	CCGCTCGAGTTATATAACAACTTCAGATTCAAAACC	
qBmAst-F	GCGTTGGTACAATTCGAGGT	qRT-PCR
qBmAst-R	TGTAAAAGCATTTTCGCTGCT	
qBmDicer2-F	GTCGATTGTCAAGTCGCTGA	
qBmDicer2-R	AAACCGGACAAGCGTGTATC	
qBmAgo2-F	GCTCCTAAAACCGAGGCTCT	
qBmAgo2-R	TAGGTTTCCTGCTGCGAGTT	
qBmActin3-F	ATCACCATCGGAAACGAAAG	
qBmActin3-R	GGTGTTGGCGTACAAGTCCT	
qBmGAPDH-F	CATTCCGCGTCCCTGTTGCTAAT	
qBmGAPDH-R	GCTGCCTCCTTGACCTTTTGC	
Vp39-F	CAACTTTTTGCGAAACGACTT	
Vp39-R	GGCTACACCTCCACTTGCTT	
dsBmAst-F	GGATCCTAATACGACTCACTATAGGGGTATCCCTGATTGGTTT	dsRNA synthesis
dsBmAst-R	GGATCCTAATACGACTCACTATAGGCTCCAGCATAATGTAGCA	
dsEGFP-F	GGATCCTAATACGACTCACTATAGGCAGTGCTTCAGCCGCTACCC	
dsEGFP-R	GGATCCTAATACGACTCACTATAGGACTCCAGCAGGACCATGTGAT	

Note: The single underlined are restriction enzyme cutting sites. The double-underlined are T7 sequences.

## Data Availability

The data presented in this study are available upon request from the corresponding author.

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
