# Peer review of "A Protein Asteroid with PIN Domain in Silkworm Bombyx mori Is Involved in Anti-BmNPV Infection"

_insects, 2023, doi:10.3390/insects14060550_

Round 1

Reviewer 1 Report

The article studied the function of BmAst in response to BmNPV infection by qRT-PCR after injection of dsBmAst. The results showed that BmAst is a constitutive gene. Knock down BmAst increased the proliferation of BmNPV significantly.  It is the first time to establish a link between nuclease and BmNPV infection in B. mori. But before acceptance for publish in the Insects, it should be carefully revised and add some content.

Major:

1.      In the introduction, there is no description of the research on the resistance of Bombyx mori against to BmNPV infection. But it is very important because this article studies a gene that related to the resistance of B. mori against BmNPV.

2.      How did authors eliminate the influence of daBmAst injection to larvae? Have you reared larvae until cocooning and pupation? And were the weights of larvae and quality of cocoons investigated?

3.      Line 221-230, how many hours after Knocking down of BmAst gene larvae were infected with BmNPV. And in the Materials and methods part there is no corresponding description. 

Minors:

1.      Line 62-63, “…. has resulted in several tools”, it is better to give some examples here.                                                                    

2.      Line 66-67, the sentence “The gene expressions of BmAst were analyzed treated by dsRNA or BmNPV” is not complete, some words such as “in the larvae of Bombyx mori” have been omitted in the middle.

3.      Line 103-105, when authors used the whole body as materials, it is better to clear that mid-gut is included or not, since it is related to the initial concentration of mRNAs.

4.      Line 124-129, the subtitle, Expressions of BmAst gene after dsRNA injection, here dsRNA is exactly dsEGFP.

5.      In line 146: Each qPCR were three replicates, here “were” should be “was”.

In the discussion part, some scientific name of the species should be italic.

    It is well written in English except some minors. 

Author Response

Comments and Suggestions for Authors

The article studied the function of BmAst in response to BmNPV infection by qRT-PCR after injection of dsBmAst. The results showed that BmAst is a constitutive gene. Knock down BmAst increased the proliferation of BmNPV significantly.  It is the first time to establish a link between nuclease and BmNPV infection in B. mori. But before acceptance for publish in the Insects, it should be carefully revised and add some content.

Response: Thank the reviewer for the valuable comments.

Major:

  1. In the introduction, there is no description of the research on the resistance of Bombyx mori against to BmNPV infection. But it is very important because this article studies a gene that related to the resistance of B. mori against BmNPV.

Response: Added the research of the resistance of silkworm against BmNPV infection.

  1. How did authors eliminate the influence of daBmAst injection to larvae? Have you reared larvae until cocooning and pupation? And were the weights of larvae and quality of cocoons investigated?

Response: We injected dsEGFP into silkworm larvae as a control to eliminate the influence of dsBmAst injection. Because of injection of dsBmAst and fed with BmNPV, the mortality of silkworm larvae was nearly 100% after 6 days injection. So we don’t reared larvae to cocooning and pupation. The weights of larvae and quality of cocoons haven’t investigated in this study.

  1. Line 221-230, how many hours after Knocking down of BmAst gene larvae were infected with BmNPV. And in the Materials and methods part there is no corresponding description.

Response: The silkworm larvae were fed with BmNPV after 48 h dsBmAst injection. And described in Materials and methods part.

Minors:

  1. Line 62-63, “…. has resulted in several tools”, it is better to give some examples here.

Response: Added.

  1. Line 66-67, the sentence “The gene expressions of BmAst were analyzed treated by dsRNA or BmNPV” is not complete, some words such as “in the larvae of Bombyx mori” have been omitted in the middle.

Response: Revised.

  1. Line 103-105, when authors used the whole body as materials, it is better to clear that mid-gut is included or not, since it is related to the initial concentration of mRNAs.

Response: The whole body included the mid gut.

  1. Line 124-129, the subtitle, Expressions of BmAst gene after dsRNA injection, here dsRNA is exactly dsEGFP.

Response: Revised.

  1. In line 146: Each qPCR were three replicates, here “were” should be “was”.

Response: Revised.

In the discussion part, some scientific name of the species should be italic.

Response: Revised through the discussion part.

Reviewer 2 Report

The study found that BmAst is highly expressed in hemocyte and fat body in silkworm, and can be induced by baculovirus but not by bacterial and fungus, and induced by dsRNA, knockdown BmAst can enhances the baculovirus infection and mortality in silkworm, suggesting that BmAst is a anti-viral gene, it is a good candidate gene for developing a baculovirus resistant silkworm.

minor issues:

line 66, The gene expressions of BmAst 66 were analyzed treated by dsRNA or BmNPV.

line 73, BmNPV T3 strain, bacterial or fungus was stored 73 at -20°C in our laboratory and purified according to the protocol reported

line 131, B. mori larvae at first day 5th were injected with dsBmAst

minor issues:

line 66, The gene expressions of BmAst 66 were analyzed treated by dsRNA or BmNPV.

line 73, BmNPV T3 strain, bacterial or fungus was stored 73 at -20°C in our laboratory and purified according to the protocol reported

line 131, B. mori larvae at first day 5th were injected with dsBmAst

Author Response

Comments and Suggestions for Authors

The study found that BmAst is highly expressed in hemocyte and fat body in silkworm, and can be induced by baculovirus but not by bacterial and fungus, and induced by dsRNA, knockdown BmAst can enhances the baculovirus infection and mortality in silkworm, suggesting that BmAst is a anti-viral gene, it is a good candidate gene for developing a baculovirus resistant silkworm.

Response: Thank the reviewer for the valuable comments.

minor issues:

line 66, The gene expressions of BmAst 66 were analyzed treated by dsRNA or BmNPV.

Response: Revised the sentence.

line 73, BmNPV T3 strain, bacterial or fungus was stored 73 at -20°C in our laboratory and purified according to the protocol reported

Response: Revised.

line 131, B. mori larvae at first day 5th were injected with dsBmAst

Response: Revised the sentence.